# ITRAQ Based Proteomics Reveals the Potential Mechanism of Placental Injury Induced by Prenatal Stress

**DOI:** 10.3390/ijms25189978

**Published:** 2024-09-16

**Authors:** Yujie Li, Junlin Hou, Liping Yang, Tong Zhang, Yu Jiang, Zhixing Du, Huizi Ma, Gai Li, Jianghui Zhu, Ping Chen

**Affiliations:** 1Department of Biochemistry, School of Medicine, Henan University of Chinese Medicine, Zhengzhou 450046, China; lyj_happy@hactcm.edu.cn; 2Department of Integrated Traditional Chinese and Western Medicine, School of Traditional Medicine, Henan University of Chinese Medicine, Zhengzhou 450046, China; houjunlin2005@163.com (J.H.); zhangtong1600@126.com (T.Z.); bio_jiangyu@126.com (Y.J.); dzx19961212@163.com (Z.D.); zydmhz@126.com (H.M.); 15225103693@163.com (G.L.); 15038268764@163.com (J.Z.); 3Gynaecology and Obstetrics, College of First Clinical Medical, Henan University of Chinese Medicine, Zhengzhou 450003, China; pinging6768@126.com

**Keywords:** prenatal stress, placental damage, proteomics analyses, PI3K/AKT/mTOR pathway

## Abstract

Maternal stress experienced during prenatal development is recognized as a significant risk factor for neurodevelopmental and neuropsychiatric disorders across the offspring’s lifespan. The placental barrier serves a crucial function in safeguarding the fetus from detrimental exposures during gestation. However, previous investigations have not yet comprehensively elucidated the extensive connections between prenatal stress and the expression of placental proteins. In this study, we used iTRAQ-based quantitative proteomics to elucidate the placental adaptive mechanisms of pregnant rats in response to fear-induced stress. Our results showed that during pregnancy, exposure to fear-induced stress led to a pathological hypercoagulable state in the mother’s body. Placental circulation was also disrupted, significantly reducing placental efficiency and blood oxygen saturation in newborn rats. Proteomic analyses showed that most of the DEPs were annotated to the PI3K-Akt and ECM-receptor interaction signaling pathway. In addition, the expressions of CDC37, HSP90β, AKT, p-AKT and p-mTOR were down-regulated significantly in the placenta. Our results demonstrated that prenatal fear-induced stress led to inhibition of the cellular signal transduction of placental PI3K/AKT/mTOR, which affected biological processes such as rRNA processing, translation, protein folding, protein stability, and oxygen transport in the placenta. These abnormalities in biological functions could potentially damage the barrier function of the placenta and thereby result in abnormal development in the offspring.

## 1. Introduction

Prenatal exposure to maternal stress is a well-established risk factor for neurodevelopmental and neuropsychiatric disorders across the lifespan of offspring. Prenatal adverse experience alters the maternal biological environment, thus directly or indirectly affecting fetal development [1,2]. Studies have shown that prenatal stress impairs the growth and development of offspring. This includes the development of the nervous system, neurocognitive functions, the immune system, the autonomic nervous system, and the hypothalamo-pituitary-adrenal (HPA)-axis [3,4]. The mechanisms of those disorders are assumed to be associated with abnormal expressions of placental receptors and enzymes, which can directly disrupt placental function, promoting alterations in the epigenetic programming of the fetus [1,5]. The placental barrier is instrumental in safeguarding the fetus against deleterious environmental exposures throughout gestation. Nonetheless, maternal chronic psychological stress can compromise this protective barrier, thereby elevating the likelihood of unfavorable developmental and health consequences [6]. The underlying molecular pathways that mediate these correlations remain poorly elucidated. Epigenetic modifications, including DNA methylation and RNA methylation, are postulated to serve as a molecular conduit connecting prenatal stress, placental barrier dysfunction, and subsequent adverse developmental and health outcomes [6]. Our prior investigative endeavors have elucidated that maternal stress induced by fear during gestation elevates the total methylation content within the placental tissue of pregnant rats. Notably, the m6A methylation modifications were predominantly localized within the coding sequences and the 3′ untranslated regions [7]. However, the implications of such epigenetic modifications on gene transcriptional activity are yet to be fully understood. Hence, the aim of the present study is to delineate the putative biological linkages between prenatal stress and placental pathological alterations through an integrative proteomic analysis. This could provide critical insights into the molecular underpinnings connecting prenatal stress, placental barrier dysfunction, and the subsequent adverse developmental and health consequences.

Our previous research has demonstrated that prenatal fear stress can precipitate depressive disorders in pregnant rats, which was found to be associated with behavioral anomalies and developmental delays in their progeny [2]. In the present study, we prepared a prenatal fear stress model (FSM) by observing and listening to the tragedy of electrically stimulated male rats according to the method described in our previous studies. Furthermore, to ascertain the model’s scientific validity and replicability, we assessed the stress response of the pregnant rats through behavioral indices and the activation state of the hypothalamic-pituitary-adrenal (HPA) axis. Concurrently, we meticulously monitored the growth and developmental trajectory of the offspring, along with their behavioral manifestations, to maintain experimental accuracy. In addition, an unbiased proteomic profiling was achieved utilizing isobaric tags for relative and absolute quantification (iTRAQ). Subsequent to data acquisition, the STRING database and Cytoscape software were instrumental in constructing protein–protein interaction networks. Moreover, Gene Ontology (GO) analyses were conducted to delineate the principal functions of the differentially expressed proteins (DEPs), while the Kyoto Encyclopedia of Genes and Genomes (KEGG) facilitated the identification of prominent pathways associated with these DEPs. Validation of the findings was accomplished through western blotting and a battery of other biochemical analyses. These results might contribute to clarifying the complex molecular mechanisms of placental dysfunction due to prenatal stress and its adverse effects on offspring development. They could potentially inform the creation of diagnostic and preventive strategies for disorders related to prenatal psychological stress.

## 2. Results

### 2.1. Behavioral Assessment of Pregnant Rats

Following a 20-day exposure to fear-induced psychological stress, pregnant rats displayed marked behavioral aberrations. The sucrose consumption of FSM rats decreased significantly, the immobility time of TST increased significantly compared with that of NC rats (*p* < 0.05 or *p* < 0.01) (Figure 1A). In addition, regarding the OFT results, the frequency of ambulation and rearing decreased significantly (*p* < 0.01) (Figure 1B). These results suggest that the exploratory behavior of rats in the FSM group was reduced, along with an increase in anhedonia and anxiety-like behaviors. Furthermore, the serum ACTH and CORT concentrations of rats in FSM group were significantly elevated compared to those of NC rats (*p* < 0.01) (Figure 1C). These results suggested that fear-induced psychological stress led to high levels of stress hormones and caused behavior deficits in pregnant rats.

### 2.2. Neurobehavioral and Developmental Milestones of the Pups

The results of the physical and neurobehavioral development of the pups are shown in Figure 1. No significant difference was observed in the average litter size between the two groups of pregnant rats (*p* > 0.05). The offspring from the FSM group (OFSM) demonstrated a pronounced extension in the duration of eye opening and incisor eruption when compared to (*p* < 0.05) those offspring from the NC group (ONC). While the ear opening, auditory startle reflex, and surface righting reflex of OFSM pups tended to be delayed in comparison to the ONC group, these differences did not reach statistical significance (*p* > 0.05) (Figure 1D). These observations suggest that maternal fear stress during gestation may impede the physical and reflexive development of the offspring.

Behavioral assessments, encompassing the SPT, TST, OFT, and MWM Test, were conducted from PND 21–27 to gauge pup anxiety levels, as well as their learning and memory capabilities. There was no significant difference in sucrose consumption between the two offspring groups (*p* > 0.05). However, OFSM rats displayed a significant increase in immobility time relative to ONC rats (*p* < 0.01) (Figure 1E). In the OFT, a significant reduction was observed in the frequency of ambulation, rearing, and self-grooming behaviors in the OFSM rats (*p* < 0.05 or *p* < 0.01) (Figure 1H). Furthermore, a statistical analysis via Repeated-measures ANOVA revealed a significant impact of training duration on the rats’ escape latency (F = 21.531, *p* < 0.01) (Figure 1F). Notably, both the ONC and OFSM groups exhibited learning progression, characterized by a continuous decrease in escape latency. However, the OFSM rats required a longer duration to reach the platform compared to the ONC rats (F = 4.694, *p* < 0.05). During the probe trial of the MWM, the OFSM rats demonstrated a significantly reduced retention time in the target quadrant compared to the ONC rats (*p* < 0.01). Despite this, no significant discrepancies were detected between the two groups in terms of swimming speed and frequency in crossing the original platform (Figure 1G). In addition, we analyzed the impact of sex on behavioral assessment results through Two-Way ANOVA. We discovered that there was a significant interaction between sex and rearing results, but no significant interaction with other behavioral results. These findings imply that maternal exposure to fear stress during gestation may augment anxiety- and depression-related behaviors, while also compromising the spatial learning and memory capabilities of the offspring.

### 2.3. The Impact of Maternal Psychological Stress on Placental Functionality and Oxygen Transfer

To evaluate placental efficiency, the weights of both the fetus and placenta were meticulously recorded. Oxygen saturation levels in the blood of both pregnant and neonatal rats were measured to ascertain the placental gas exchange capabilities. As shown in Figure 2A–D, exposure to fear stress during pregnancy significantly diminished fetal and placental weights, as well as the efficiency of the placenta, when contrasted with the non-stressed control group (*p* < 0.01). Moreover, the blood oxygen saturation in the FSM group was markedly lower compared to that of the control rats (Figure 2E). Additionally, the mean corpuscular volume (MCV) witnessed a pronounced reduction (*p* < 0.01) (Table 1), whereas there was a significant increase in platelet count and mean platelet volume within the FSM group. Furthermore, compared with NC rats, the APTT, TT, and the ATIII activation of FSM rats were significantly decreased (*p* < 0.01 or *p* < 0.05) (Figure 2G,H). These results suggested that maternal fear stress during gestation may elicit a hypercoagulable state, which potentially disrupts placental circulation and results in abnormalities in embryonic development.

### 2.4. Effects of Prenatal Fear Stress on Placental Histomorphology in Rats

In the initial assessment, the gross morphological characteristics of placental tissue were examined. We noted that the NC group exhibited a lighter, cherry-red hue, in contrast to the FSM group, which demonstrated a more intense, purplish-red hue (Figure 2A). Furthermore, a marked decrease in both the diameter and the thickness of the placenta was noted in the FSM group when compared with the NC group (Figure 2B,C). Histological evaluations, including H&E staining and Masson staining, were conducted to evaluate the placental histopathological changes (Figure 2J). Our analysis revealed a diminished vascular network within the placentas of the FSM group (Figure 2I), accompanied by an increased number of red blood cells in the vascular channels observed under the microscope (Figure 2F). Those results might be related to the pathological hypercoagulable state of pregnant rats.

### 2.5. iTRAQ-Based Proteomic Analyses

The placenta plays an important role in gas exchange and nutrient transport between mother and fetus. Maternal psychological stress has the potential to perturb these interactions, subsequently impacting the somatic and neural development of the fetus [8]. To delineate the influence of maternal psychological stress on placental functionality, we conducted a proteomic study of placental proteins using iTRAQ coupled with LC-MS/MS techniques. In doing so, we identified 1753 proteins from 16,019 distinct peptides derived from 339,465 spectra with a high confidence (≥95% confidence) and unique peptide matches ≥ 2. After excluding uncharacterized proteins and duplicate values caused by different peptide segments of the same protein, a total of 1730 proteins were identified. Using fold changes > 1.2 or <0.83 with *p*-value < 0.05 as a cutoff, 287 DEPs were identified between the two groups, including 145 upregulated proteins (Appendix B Table A1) and 142 downregulated proteins (Appendix B Table A2). The top 10 up- and down-regulated proteins in the placenta are shown in Table 2. Figure 3A depicts a quantitative volcanic map of differentially expressed proteins. 

### 2.6. Protein–Protein Interaction (PPI) Network Construction and Hub Genes Extraction

The PPI network of these DEPs was performed by STRING software (STRING Database Version: 12.0), as presented in Figure 4. There were 170 nodes and 575 edges in the network. It consisted of 1 major network and 12 minor networks (Figure 4A). Utilizing the cytoHubba plug-in of Cytoscape software, hub genes were extracted from the PPI network (Figure 4B). A total of 39 hub genes were identified using eight different algorithms, encompassing 21 ribosomal proteins (involved in the translation process) and 18 additional proteins. These hub genes are predominantly involved in transcriptional regulation (Snrpd3), translation (21 ribosomal proteins, Eef2, and Eef1a1), protein folding and refolding (Hsp90ab1, Hspa1a, Tcp1, Cct5), and coagulation (C4bpa, Fgg, Serpine1) [9,10,11,12] during the critical phase of placental development.

### 2.7. GO and KEGG Enrichment Analyses of DEPs

In pursuit of elucidating the impact of maternal psychological stress on placental functionality, GO annotation (Figure 3B–D) and KEGG pathway enrichment (Figure 5A) were conducted on the DEPs. The results indicated that fear stress during pregnancy significantly affects a diverse array of placental biological processes, encompassing blood coagulation, fibrin clot formation, Fibrinolysis, negative regulation of endopeptidase activity, rRNA processing, protein folding, protein stabilization, translation, oxygen transport, and other biological processes. The molecular function enrichment of DEPs was predominantly concentrated in protein binding, RNA binding, and structural constituent of ribosome and oxygen binding. According to the GO cellular component enrichment, the DEPs were predominantly situated within the cytosolic ribosome, basement membrane and cytoplasm. Pathway enrichment analysis highlighted the ECM-receptor interaction and the PI3K-Akt signaling pathway as the primary pathways associated with the DEPs. 

To delineate the precise contributions of ECM-receptor interaction and PI3K-Akt signaling pathway in the context of placental damage induced by fear stress, Cytoscape software was utilized to construct a comprehensive correlation network integrating the insights from pathway enrichment with the identified differential proteins (Figure 5B). The analysis suggested that the PI3K-Akt signaling pathway intersects with several pathways, including protein digestion and absorption, small cell lung cancer, focal adhesion, ECM-receptor interaction, proteoglycans in cancer and amoebiasis pathways. The results indicate that maternal psychological stress has the potential to perturb the PI3K-Akt signaling pathway, compromising the placenta’s normal physiological functions, thereby impacting fetal physiological and neural development.

### 2.8. The Impact of Maternal Stress on Placental PI3K/AKT/mTOR Pathway Signaling

As shown in Figure 6, to explore the potential mechanisms of placental injury induced by fear stress, we conducted an analysis of placental protein expression, focusing on PI3K, p-PI3K AKT, p-AKT, mTOR, and p-mTOR expression. Compared with the NC group, the expressions of PI3K and mTOR proteins were without significant changes (*p* > 0.05). However, the expressions of AKT, p-AKT, and p-mTOR decreased significantly. Therefore, fear stress may affect placental development via inhibiting cell signaling of PI3K/AKT/mTOR, without influencing the expressions of PI3K and mTOR.

### 2.9. The Impact of Maternal Stress on Placental Cdc37 and HSP90β Protein Expression

The pathway analysis showed that the expressions of CDC37 and HSP90β proteins in PI3K-Akt signaling pathway decreased significantly in the FSM group. The western blotting assay was employed to verify these results, as shown in Figure 6. The expression of placental Cdc37 (Figure 6J,K) and HSP90β chaperones (Figure 6J,L) were significantly decreased in the group exposed to prenatal stress. These results were consistent with the proteome detection outcomes. Our results suggest that the decreased expressions of Cdc37 and HSP90β chaperones in the placenta may affect placental development via inhibiting cell signaling of PI3K/AKT.

In conclusion, we demonstrated that prenatal fear stress induced decreases in the expressions of Cdc37 and HSP90β chaperones in the placenta, further inhibiting the PI3K/AKT/mTOR pathway (Figure 7). This disruption in placental signaling is likely to impact fetal growth and development.

## 3. Discussion

We investigated the effects of chronic fear stress on placental proteins expression in pregnant rats, iTRAQ-based proteomics analysis was employed for obtaining the unbiased profiling data. These findings are instrumental in elucidating the mechanisms underlying placental injury caused by chronic fear stress in pregnant rodents. Our previous research has demonstrated that such stress not only precipitates emotional disturbances in the dams but also impairs the growth, development, and cognitive abilities of their offspring [2]. Similar results have been reported by Brannigan et al. [13], who observed that offspring exposed to maternal stress during gestation exhibited an elevated predisposition to psychiatric conditions in later life. This mechanism is attributed to elevated corticosterone levels in the maternal plasma. Chronic exposure of the placenta to heightened glucocorticoid concentrations results in diminished 11β-HSD2 activity, thereby facilitating a substantial increase in corticosterone levels reaching the fetal circulation, which in turn affects fetal development [14]. The placental barrier is crucial for safeguarding the fetus from deleterious exposures throughout pregnancy. Our initial investigations demonstrated that stress induced by fear during gestation elicits modifications in m6A methylation within the mother’s placental tissue [7]. However, the specific proteins within the placenta that may undergo abnormal expression due to such alterations remain undefined. To date, prior research has yet to provide an exhaustive comprehension of the extensive interconnection between prenatal stress and placental protein expression. Proteomic inquiries are imperative to deciphering the potential biological correlations between prenatal stress and pathological alterations in the placenta.

In this study, pregnant rats were subjected to prolonged fear-induced stress throughout their gestational period by observing the application of electrical stimulation to male counterparts. Three different stress-response phenotypes and the levels of stress hormones of the prenatal fear stress rat model were evaluated. The results demonstrated that the rats that were subjected to fear-induced psychological stress manifested anhedonia-like behavior, evidenced by a diminished preference for sucrose water compared to the control rats. Moreover, these rats exhibited increased immobility during the TST and a reduced propensity for exploration in the OFT, indicative of behavioral depression. Additionally, elevated serum stress hormone concentrations confirmed the stress condition in the pregnant rats. Collectively, the prenatal fear stress rat model, induced by the observation of electrical stimulation in male rats, serves as an appropriate animal model for studying psychological stress during pregnancy. To investigate the correlation between prenatal fear stress and placental damage, as well as to elucidate the detrimental impacts of such injuries on offspring development, we conducted assessments of the progeny’s behavioral patterns. Our investigation revealed that progeny of rats subjected to fear stress manifested growth retardation, accompanied by deficits in learning and memory functions. These outcomes were consistent with our previous reports [2], thereby affirming the consistency and reliability of our methodology. A significant benefit of this model is that it is an exclusively psychological stress model. The stressor itself does not result in severe physical harm to the pregnant rats, such as exposure to humidity, starvation, pain, or other adverse stimuli that could potentially influence fetal development. Consequently, the psychological stress model employed in our study is an optimal choice for assessing maternal stress during pregnancy, offering critical insights into the pathogenesis of Emotion-Caused Disease Theory during this critical period.

Placental tissue is integral to the oxygen and nutrient transfer between mother and fetus, facilitating essential maternal–fetal interactions. The functions of the placenta extend beyond its primary role, encompassing scavenging and endocrine activities. It secretes neuroactive signaling molecules with endocrine, paracrine, and autocrine effects, regulating the activity of other endocrine glands and thereby influencing maternal adaptation to pregnancy [15]. As the physiological conduit between mother and fetus, the placenta serves as a critical barrier against harmful prenatal exposures. Nevertheless, psychological stress during pregnancy may compromise these protective mechanisms, potentially impairing fetal physiological and neural development [8]. Our results illustrated that fear stress in pregnant rats during the gestation period might induce a pathological hypercoagulable state and reduce blood oxygen saturation. These hematological pathological changes in pregnant rats would impact the placental O2 and nutrient transfer at the maternal–fetal interface, and ultimately lead to abnormal embryonic development. Normal pregnancy is characterized by a physiological hypercoagulable state, which serves a protective role in mitigating bleeding complications during parturition [16]. Exaggerated hematological pathological changes in pregnant rats may lead to placental ischemia, subsequently impacting the gas exchange and nutrient transport between mother and fetus [17]. Chronic stressors, such as occupational strain, posttraumatic stress disorder, and psychological distress from depressive and anxiety symptoms, activate the sympathoadrenal medullary system, resulting in a chronic low-grade hypercoagulable state [18,19]. In the context of pregnancy, when such stressors arise, the hematological pathological alterations are magnified in pregnant rats, potentially precipitating placental ischemia. This condition may result in compromised maternal–fetal circulation, thereby predisposing the fetus to preeclampsia, fetal growth restriction, abruptio placentae, intrauterine death, and stillbirth [20,21,22]. These results indicate that maternal exposure to fear stress throughout pregnancy may elicit a pathological hypercoagulable condition in pregnant rats, potentially diminishing placental functionality and ultimately impacting the ontogenetic progression of the embryos. The results may help with the development of strategies for the diagnosis and prevention of diseases caused by prenatal psychological stress.

The atypical hypercoagulable condition not only impacts the interchange of gases and nutritional substances between the mother and the fetus, but also alters the structural integrity of the placenta itself [23]. Consequently, we conducted morphological assessments of the placenta utilizing H&E staining and Masson staining techniques. Additionally, we evaluated fetal weight, placental weight, and placental efficiency as indicators of placental anomalies. Our morphological findings revealed an accumulation of red blood cells within the placental vasculature in the FSM group, a condition potentially linked to the pathological hypercoagulability observed in the pregnant rats. Furthermore, there was a significant reduction in fetal weight, placental weight, and placental efficiency observed in rats subjected to psychological stress. These findings corroborate our hypothesis that chronic fear stress during pregnancy induces pathological hypercoagulability in pregnant rats, thereby mediating placental ischemia and inflicting substantial harm to neonates.

To elucidate the impact of maternal psychological stress on placental functionality, our research entailed a comprehensive proteomic examination of placental tissue to discern the biological underpinnings associating stress with pathological changes in the placenta. The results revealed that 287 proteins exhibited marked differential expression between the two groups. When we entered the glucocorticoid receptor gene NR3C1 along with 287 differentially expressed proteins into the STRING database for protein interaction analysis (STRING score = 0.4, FDR stringency = 0.01), it was discovered that NR3C1 can interact with two of these genes (stip1 and ptges3), suggesting that their expression may be modulated by the glucocorticoid receptor signaling pathway. Bioinformatics analysis revealed that these differential proteins were mainly enriched in blood coagulation, coagulation, fibrin clot formation, rRNA processing, translation, protein folding, protein stabilization, oxygen transport, and other biological processes of the placenta. The molecular functions of the DEPs were mainly involved in protein binding, RNA binding, and a structural constituent of the ribosome. The pathway enrichment analyses of the DEPs were mainly involved in ECM-receptor interaction and PI3K-Akt signaling pathway. PPI analysis revealed that 39 hub genes may play a significant role in the impairment of placental function due to pregnancy-related stress. Subsequent analysis indicated that among these genes Fgg, Alb, Vtn, Gnaq, Tcp1, cct5, Hsp90ab1, and eEF2 were regulated by the PI3K-Akt signaling pathway.

Phosphoinositide 3 kinase (PI3K) serves as a pivotal molecule in cellular signal transduction, with its principal downstream effector being the serine/threonine-protein kinase, protein kinase B (AKT) [24]. PI3K exhibits dual enzymatic activities, functioning as both a lipid kinase and a protein kinase. Activation of the PI3K signaling pathway leads to an increase in the production of PIP3, which subsequently associates with the intracellular signaling protein AKT, bearing a PH2 domain, and elevates its phosphorylation status. Phosphorylated AKT (p-AKT) is capable of phosphorylating the tuberous sclerosis complex (TSC1/2), thereby facilitating the release of RHEB and the subsequent activation of the mammalian target of rapamycin (mTOR) [25]. mTOR, a key cellular signaling molecule, influences cytokine expression, transcription, and protein synthesis, and governs cellular processes such as growth, autophagy, and apoptosis. As a serine/threonine protein kinase, mTOR can either complex with the rapamycin-associated TOR (RAPTOR) protein to form mTOR complex 1 (mTORC1) or with the rapamycin-insensitive companion of mTOR (RICTOR) to form mTOR complex 2 (mTORC2). Conditions of hypoxia or energy deficiency can result from the inhibition of mTOR and the suppression of protein translation. In mice deficient in either mTOR or RAPTOR, fetal demise occurs around the peri-implantation period, highlighting the crucial role of mTOR in early placental and embryonic development [26]. Fetal growth is intimately associated with modifications in the placental mechanistic targeting of mTOR signaling, with reduced signaling observed in fetal growth restriction and increased signaling in fetal overgrowth [27]. It has been postulated that the placenta orchestrates a complex integration of maternal and fetal nutritional signals with intrinsic nutrient-sensing pathways to align fetal demand with maternal supply, regulating maternal physiology, placental growth, and nutrient transport, with trophoblast mTOR playing a central role in this homeostatic regulatory mechanism [28]. In the occurrence of an unusual insufficiency in uteroplacental blood perfusion, the placenta predominantly exhibits a suppression of both the mTORC1 and mTORC2 signaling axes. Following this suppression, there is a decrement in the placental nutrient transport mechanisms, as evidenced by the diminished activity of amino acid and folate transporters. Concurrently, this is associated with a decline in mitochondrial activity and protein biosynthesis, cumulatively leading to a reduced fetal nutrient supply and ultimately culminating in intrauterine growth restriction (IUGR) [28]. Furthermore, the administration of corticosterone to pregnant rodents has been documented to suppress placental mTORC1, mTORC2, p-4EBP1, and p-Akt [29]. In this study, the protein expression levels of AKT, p-AKT, and p-mTOR in placental tissue were significantly reduced in the FSM group compared with those in the NC samples. These results lead us to speculate that placental PI3K/Akt/mTOR signaling pathway dysfunction plays a vital role in the process of placental dysfunction caused by maternal psychological stress. 

In the context of untreated cellular environments, a significant proportion—approximately 30%—of Akt is found to be associated with a complex involving HSP90 and cdc37. This association is potentially pivotal for the maintenance of Akt stability, as indicated by reference [30]. The inhibition of cdc37 has been observed to weaken the PI3K/AKT kinase signaling pathways, which are integral to the regulation of the trophoblast cell cycle and proliferation, as documented in references [31,32]. The Hsp90 protein, encoded by the genes hsp90aa1 and hsp90ab1, corresponds to the Hsp90α and Hsp90β subtypes, respectively. Notably, the expression of Hsp90β exhibits a constitutive pattern, whereas Hsp90α expression is significantly elevated in response to cellular stress, as per reference [33]. Hsp90β is essential for mouse development, while Hsp90α plays a role in maintaining male fertility in adult mice, as stated in reference [34]. In this study, the expression levels of Cdc37 and HSP90β chaperones in placental tissue were significantly reduced in the FSM group when compared to those in the NC samples. These findings were in line with the proteome detection results. Our results imply that decreased expressions of Cdc37 and HSP90β chaperones in the placenta might impact placental development by inhibiting the cell signaling of PI3K/AKT.

In conclusion, we employed LC-MS and iTRAQ based proteomics to ascertain if fear stress in pregnant rats has an impact on placental proteomics. We discovered that fear stress during pregnancy elicited elevated glucocorticoid concentrations and pathological hypercoagulability, ultimately leading to the suppression of the placental PI3K-Akt signaling pathway. This disruption impacts a spectrum of placental biological processes, including rRNA processing, translation, protein folding, protein stability, and oxygen transport. Such dysfunctions may compromise the placental barrier function, potentially resulting in developmental abnormalities in the offspring.

## 4. Materials and Methods

### 4.1. Animal Care and Ethics Statement

A total of 30 female Wistar rats and 15 male Wistar rats aged 11 weeks and weighing 230–260 g [Purchased from Beijing weitonglihua Experimental Animal Technology Co., Ltd., Beijing, China, license number: SCXK (Jing) 2016–0006] were housed in standard laboratory conditions (20 ± 3 °C, 45 ± 5% relative humidity, 12/12 h light/dark cycle) and were given access to food and water ad libitum. All experiments were approved by the Ethics Committee of Henan University of Chinese Medicine (Zhengzhou, China) under the regulations of National Institutes of Health guidelines for the care and use of laboratory animals.

### 4.2. Animal Treatment

#### 4.2.1. Prenatal Fear Stress

All female rats and male rats were mated in a cage with the ratio of 2:1 according to our previous study [2]. Mating and gestation were estimated based on the presence of sperm in the vagina of female rats. After fertilization, 24 pregnant rats were randomly assigned to either the normal control (NC) or fear stress model (FSM) group, with 12 rats in each. The prenatal fear stress procedure was conducted according to our previous study [2]. Briefly, FSM pregnant rats were placed into an improved electric shock box to observe and listen to the traumatic event of male rats being subjected to electric stimulations for 20 days (Figure 8) [35,36,37]. Control pregnant rats were placed in the same situation but did not undergo the psychological fear stress procedure. In the NC group, the pregnant female rats were subjected to identical conditions; however, no galvanic stimulation was applied to the base of the apparatus for the box. Consequently, the male rats were not exposed to electrical currents and, as a result, did not manifest behaviors such as screaming or jumping, thereby failing to communicate any fear-related cues. In periods of fear stress, the pregnant female rats were raised in separate cages from the male rats.

#### 4.2.2. Stress State Assessment of Pregnant Rats

In the final week of gestation, the sucrose preference test (SPT), open field tests (OFT), and tail suspension test (TST) were administered to evaluate anxiety profiles in the pregnant rats [37,38]. Concurrently, serum ACTH and CORT concentrations were quantified on the 20th day of pregnancy using enzyme linked immunosorbent assay (ELISA) to assess stress hormone responses.

#### 4.2.3. The Behavioral Assessment of the Offspring

Four pregnant rats gave birth naturally in each group, and 39 and 32 neonatal rats were obtained in the NC group and FSM group, respectively. The behavioral and developmental assessment of neonatal progeny was conducted according to our previous study (see Figure 9). Specifically, the chronological milestones of physical maturation of all pups in each dam, including the acquisition of surface righting reflexes and auditory startle reflexes, were meticulously documented. Furthermore, the onset of eye opening, incisor emergence, and auricular patency were meticulously recorded to ascertain the pups’ physiological development and maturation. In addition, between postnatal days (PNDs) 21 and 27, 10 offspring rats (5 males and 5 females) were selected from each group for behavioral assessment; the SPT (PND 21–23), OFT (PND 21 a.m.), the Morris Water Maze (MWM, PND 23–27), and the TST (PND 21 p.m.) were administered to evaluate the offspring’s anxiety levels and their cognitive learning and memory abilities.

### 4.3. Method of Behavioral Testing

#### 4.3.1. Open Field-Test

The open field-test was conducted as follows [39]: rats were positioned in the central area of an open-field apparatus measuring 100 cm × 100 cm × 50 cm. The apparatus floor was segmented into 25 equivalent sections. The movement tracks of the rats were recorded via a video camera over a span of 5 min, with the apparatus being sanitized with alcohol between each trial.

#### 4.3.2. TST

The TST was performed in accordance with established methodologies [40]. Rats were hung by their tails for a period of 6 minutes. Immobility was defined as a complete lack of movement. The cumulative immobility time was recorded for the final 4 min of the 6-min session.

#### 4.3.3. SPT

The SPT [40] was conducted as follows: Initially, each rat was individually housed in a single-cage environment, and acclimated to a 1% sucrose solution for the duration of 24 h. Subsequently, they underwent a 12-h fast from both food and water. On the third day of the experimental protocol, pairs of identical bottles containing 1% sucrose solution and tap water were placed in each cage concurrently. After an hour, the positions of the two bottles were interchanged. One hour later, the remaining volumes of the liquids were quantified. The percentage of sucrose preference was measured as follows: sucrose preference = sucrose consumed/(sucrose consumed + tap water consumed).

#### 4.3.4. MWM

The MWM Test was performed according to a previously reported protocol [41]: The water maze was a black circular pool (100 cm in diameter, 60 cm in height) with a stable platform positioned 1 cm beneath the water surface. Prior to testing, the maze was filled with tap water at a temperature of 25–27 °C and rendered opaque by the addition of a non-toxic white paint. The experiment comprised two sequential phases: place navigation (initial training) and spatial probe (the Space Exploration Test). During the initial training phase, the platform was situated centrally in the north quadrant. Each rat was placed in the pool, oriented towards the wall, at one of four starting points (north, south, east, and west) in a semirandom sequence, and allowed 120 s to locate the submerged platform. Rats that voluntarily found the platform were allowed a 5-s stay, whereas those that failed to do so within the allocated time were gently directed to the platform and held there for 30 s. Each rat underwent four trials daily over a period of 4 consecutive days, the time spent by the rats to reach the platform (escape latency times) were recorded. Following the 4-day training regimen, the spatial probe test was performed to evaluate memory retention. On the 5th day, the platform was removed, and each rat was released into the water from the quadrant opposite to the platform’s previous location, with a 90-s search period. The frequency of each rat’s crossings over the original platform position, swimming velocities, and the duration spent in the target quadrant were meticulously documented and analyzed using a video-tracking system (CG-400 Image Acquisition System; Institute of Materia Medica, Chinese Academy of Medical Sciences, Shanghai, China). Post-trial, the rats were dried and maintained at a comfortable temperature before being returned to their enclosures.

### 4.4. Sample Preparation

On the 20th day of pregnancy, six pregnant rats in each group, after a 12-h fast, were subjected to fear stress for 30 min daily, followed by immediate anesthesia with pentobarbital sodium (50 mg/kg, intraperitoneally). Blood samples were collected from the abdominal aorta; the K2-EDTA anticoagulated whole blood, sodium citrate anticoagulated plasma, and serum were collected. All blood samples were procured within a 2-h window following the rats’ exposure to fear stress. Sodium citrate anticoagulated plasma and serum were stored frozen, and all biochemical tests were completed within 1 month. The fetuses and placentas were swiftly separated, and the number of litters was recorded, and then weighed using a digital scale. Additionally, the diameters and thicknesses of the placentas were manually measured using a caliper (Mitutoyo, Kanagawa, Japan). Placental efficiency was expressed as the ratio of fetal weight to placental weight. Three placentas were randomly selected from each dam, and stored in liquid nitrogen, used for Western blot analysis and proteomic analysis. Each placenta was randomly selected from the respective dams and subsequently immersed in a 4% paraformaldehyde solution at a temperature of 4 °C for a duration of one week to facilitate H&E staining procedures. In the experimental cohort, the remaining pregnant rats underwent spontaneous parturition. On the 20th day of gestation, the blood oxygen saturation levels in these rats were assessed. Furthermore, the oxygen saturation in their progeny was quantified on PND 1 and PND 21 utilizing a PC100SEV animal oximeter (Wuhan Weisi Biotechnology Co., Ltd., located in Wuhan, China).

### 4.5. Biochemical Assay

Utilizing K2-EDTA anticoagulated whole blood, complete blood count (CBC) determinations were performed on the day of sample collection via an Mairui animal automatic blood cell analyzer (BC-2800vet, Shenzhen, China). Concurrently, plasma coagulation factors were assayed using in vitro diagnostic kits (cat. no. R01002, R01102, R01202, and R01302, Redu Life Sciences Co., Ltd., Shenzhen, China). Additionally, serum D-dimer and Antithrombin-III levels were quantified by enzyme linked immunosorbent assay (cat. no. JL21114, JL15661, Shanghai Jianglai Biotechnology Co., Ltd., Shanghai, China), along with ACTH and CORT concentration measurements (cat. no. H097-1-2, H205 Shanghai Jianglai Biotechnology Co., Ltd., Shanghai, China) to assess stress hormone responses and the functionality of both coagulation and anticoagulant systems.

### 4.6. Histological Examination

The placental tissues within a 3 mm radius from the insertion point of the umbilical cord were selected. The placenta tissues were dehydrated with graded alcohol series (80%, 95%, and 100%) and embedded in paraffin. They were then sliced into serial transverse 4-μm-thick sections, six slices for each sample. The sections were stained with hematoxylin & eosin (H&E) and Ponceau-s & aniline blue (Masson).

### 4.7. Proteomic Analysis

#### 4.7.1. Protein Extraction

Placental samples with a diameter of 5 mm around the umbilical cord were isolated, approximately 100 mg of the placental sample from each rat was homogenized in liquid nitrogen and mixed well with 6 × volume of chilled TCA-acetone (500 mL acetone, 50 g TCA), and precipitated at −20 °C for about 2 h. After centrifugation at 4 °C and 20,000× *g* for 30 min, the supernatant was discarded, and the precipitate was mixed well with 4 volume equivalents of chilled acetone, precipitated at −20 °C for 30 min, centrifuged at 4 °C and 20,000× *g* for 30 min again. The precipitate was dissolved with lysis buffer (8 M urea, 30 mM HEPES, 1 mM PMSF, 2 mM EDTA, 10 mM DTT) and sonicated at 180 W for 5 min (pulse on 2 s pulse off 3 s for each time). The mixture was centrifuged at 4 °C and 20,000× *g* for 30 min again, the supernatant was collected and added to DTT (10 mM), the mixture was incubated at 56 °C for 1 h, then added to IAM (55 mM) and incubated in darkness for 1 h, then 6 × volume of chilled acetone was added and precipitated at −20 °C for at least 3 h. Having subsequently undergone centrifugation at 4 °C and 20,000× *g* for 20 min, the precipitate was collected and redissolved in 700 µL of dissolution buffer (50% TEAB, 0.1% SDS) and sonicated at 180 W for 3 min. After centrifugation at 4 °C and 20,000× *g* for 30 min, the supernatant was collected and the protein concentrations were quantified using the Bradford method (Pierce™ Coomassie [Bradford] Protein Assay Kit, ThermoFisher, Waltham, MA, USA).

#### 4.7.2. Protein Digestion and Peptide iTRAQ Labeling

Proteins (100 µg per sample) were added to the dissolution buffer (50% TEAB, 0.1% SDS) and digested with trypsin. This was then freeze-dried and redissolved with 30 µL dissolution buffer (50% TEAB, 0.1% SDS). The tryptic peptides were labeled with an iTRAQ^®^Reagent-8 Plex Multiplex Kit (Applied Biosystems, Waltham, MA, USA), following the manufacturer’s instruction. The samples of NC group were labeled with iTRAQ reagent 113, 114, 115, 116, and the samples of FSM group were labeled with iTRAQ reagent 117, 118, 119, 121, respectively (Table 3). The labeled samples were incubated at room temperature for 2 h, the reaction was terminated by the addition of 10 × volume of dilution [10 mM KH2PO4, 25% acetonitrile (ACN), pH = 3.0], centrifuged at 4 °C and 15,000× *g* for 10 min, the supernatants were then collected and pooled. The peptides were preliminarily separated by strong cation exchange (SCX) chromatographic-HPLC. The chromatography conditions were: the labeled peptides were loaded onto an SCX chromatographic column (Luna SCX 250 × 4.60 mm 100 Å, phenomenex) and equilibrated in 100% solution A for 10 min, followed by a fast elution. The elution reagents consisted of the solvent A (10 mM KH2PO4 in 25% ACN, pH = 3.0) and solvent B (2 M KCL, 10 mM KH2PO4 in 25% ACN, pH = 3.0, flow rate  =  1 mL/min). The gradients were as follows: 0% B for 45 min; 0–5% B for 1 min; 5–30% B for 10 min; 30–50% B for 5 min; 50% B for 5 min; 50–100% B for 5 min; and 100% B for 10 min. After 46 min of elution, the fractions were collected at a rate of 1 tube/min, and the fractions with few peaks were combined. The fractions were desalted using a C18 column (strata-X C18, phenomenex), lyophilized, and dissolved in 0.1% formic acid for LC-MS/MS analyses.

#### 4.7.3. LC-MS/MS Analyses

The peptides in each fraction were loaded onto an Acclaim PePmap C18-reversed phase column (3 µm, 75 µm × 2 cm 100 Å, Thermo Scientific, Waltham, MA, USA) and connected with a reversed phase C18 column (5 µm, 75 µm × 10 cm 300 Å, Agela Technologies, Tianjin, China) for separation, and then gradiently eluted with 5–80% (*v*/*v*) ACN in 0.1% formic acid for 65 min at 300 nL/min. The peptide fractions were analyzed using a Q Exactive mass spectrometer (Thermo Fisher Scientific, MA, USA). The detection method was positive ion, the capillary voltage was 1800 V, and the capillary temperature was 300 °C. The acquisition ranges were 350–2000 *m*/*z* at a resolution of 70,000 for the MS. The resolution for MS/MS scan was set to 17,500 with a minimum signal threshold at 1 × 10^5^ and isolation width at 2 *m*/*z*.

#### 4.7.4. Protein Identification and Quantification

The identification and quantification of peptides and proteins were constituted using the ThermoFisher Proteome Discoverer (version 1.3) tool with Sequest HT search engine based on the UniProt rat database. The user-defined search parameters were set as follows: trypsin; missed cleavages, two; fixed modifications, iTRAQ 8plex on N-term and Lys, and Carbamidomethylation on Cys; variable modification, Oxidation on Met, deamidation on Asn and Gln, and Pyro-Glu, and acetylation on protein N-term; precursor mass tolerance of 15 ppm; fragment mass tolerance of 0.05 Da. The false discovery rate for peptide spectrum matching in protein and peptide identification with the filter module is below 0.01, necessitating further data screening. The ion quantifiers, along with peptide and protein quantifier nodes, were employed to compute and quantify the relative proportions of peptides and proteins in the sample. Each identified protein involved at least one unique peptide. After these procedures, we obtained the quantitative results of protein expression data. After weighing and normalizing by the median ratio, the resultant ratio data were subjected to statistical evaluation via a two-tailed Student’s *t*-test to verify the significance of the differences between the two groups: the NC group and the FSM group. Proteins with a change greater than 1.2-fold and a *p*-value lower than 0.05 were regarded as DEPs.

#### 4.7.5. Bioinformatics Analysis

The protein–protein interaction network was constructed by the STRING database [42]. The functional annotation of DEPs was performed using the Gene Ontology (GO) database. The prediction of the significant pathways was carried out with Kyoto Encyclopedia of Genes and Genomes (KEGG) database [43,44,45]. Cytoscape 3.7.2 software was used to visualize the results.

### 4.8. Western Blot Analysis

The total proteins from the placenta was extracted using RIPA lysis buffer (50 mM Tris [pH 7.4], 150 mM NaCl, 1% Triton X-100, 1% sodium deoxycholate, 0.1% SDS) with PMSF, protease inhibitor and a phosphatase inhibitor. A BCA protein quantitative detection kit (Jiangsu Meimian Industrial Co., Ltd., Changzhou, China) was used to determine the protein content. The protein was separated with SDS-PAGE and transferred to PVDF membranes with electrophoresis systems (Bio-Rad 1645050, Hercules, CA, USA). The PVDF membranes were blocked with 5% (*w*/*v*) skimmed milk powder for 2 h and incubated at 4 °C overnight with the following primary antibodies anti-PI3K (Cat No.GB11525-100, Servicebio, Wuhan, China), anti-AKT (Cat No.GB15689-100, Servicebio, Wuhan, China), anti-phospho-AKT (p-AKT, Cat No.GB 150002-100, Servicebio, Wuhan, China), anti-mTOR (Cat No.GB111840-100, Servicebio, Wuhan, China), Phospho-mTOR (p-mTOR, Cat No.67778-1-ig, Proteintech, Wuhan, China), anti-CDC37 (Cat No.66420-1-ig, Proteintech, Wuhan, China) anti-HSP90β (Cat No.GB111280, Servicebio, Wuhan, China), and anti-α-tubulin (Cat No.GB15201-100, Servicebio, Wuhan, China). After being washed with 1× TBST solution (Servicebio, Wuhan, China) three times, the membranes were incubated with HRP-labelled secondary antibodies (Servicebio, Wuhan, China). A gel imaging system (Thermo Fisher Scientific, MA, USA) and Image J Software were used for imaging and statistical analysis. α-tubulin was used as an internal control to ensure equal protein loading.

### 4.9. Statistical Analyses

Measurement data was presented as means ± SDs, and the statistical analyses were performed using SPSS 21.0 statistical software (IBM, Armonk, NY, USA). The escape latency of offspring was analyzed using repeated-measures ANOVA. The offspring’s other behavioral results were analyzed through Two-Way ANOVA. The other measurement data were analyzed using independent *t*-test. *p* < 0.05 was considered statistically significant.

## 5. Conclusions

Prenatal fear stress induced the inhibition of the cellular signal transduction of placental PI3K/AKT/mTOR, which affected biological processes such as rRNA processing, translation, protein folding, protein stability, and oxygen transport in the placenta. The results suggested that fear stress during pregnancy elicited elevated glucocorticoid concentrations and pathological hypercoagulability, ultimately leading to the suppression of the placental PI3K-Akt signaling pathway. This disruption impacts a spectrum of placental biological processes, including rRNA processing, translation, protein folding, protein stability, and oxygen transport. Such dysfunctions may compromise the placental barrier function, potentially resulting in developmental abnormalities in the offspring.

## Figures and Tables

**Figure 1 ijms-25-09978-f001:**
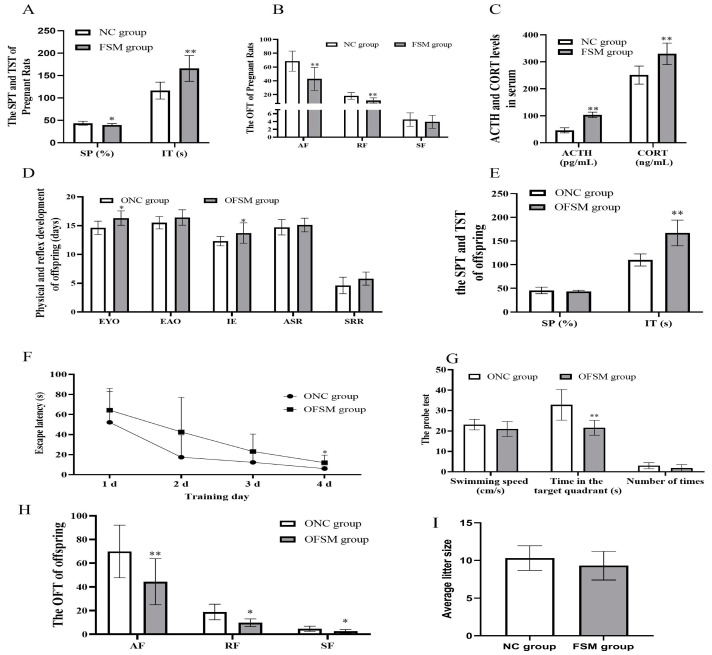
Behavioral assessment of pregnant rats and their pups. The results of (**A**) the Sucrose Preference Test and the immobility time of pregnant rats are presented. (**B**) The results of Open-Field Test of pregnant rats. (**C**) The concentrations of serum stress hormones in pregnant rats. (**D**) The physical and reflex development of offspring. (**E**) The sucrose consumption and the immobility times in the Tail suspension test of pups. (**F**) The escape latency during the 4 training days in the MWM and (**G**) the swimming speeds, the retention time that rats spent in the target quadrant, and the number of times that the pups crossed the original platform during the probe test in the MWM. (**H**) The frequency of ambulation, rearing, and self-grooming in the open field-tests of the pups. (**I**) The result of the average litter size of pregnant rats. NC group, normal control group; FSM group, fear stress model group; SP, Sucrose Preference; IT, immobility time; AF, Ambulation frequency: number of floor units entered with all four feet; RF, Rearing frequency, number of instances of standing on the hindlimbs without touching the wall; SF, Self-grooming frequency: number of self-grooming actions performed; ACTH, Adrenocorticotropic hormone; CORT, Corticosterone; OFSM, offspring from the FSM group; ONC, offspring from the normal control group; EYO, eye opening; EAO, ear opening; IE, incisor eruption; ASR, auditory startle reflex; SRR, surface righting reflex; MWM, Morris water maze. Values are expressed as means ± standard deviations.* *p* < 0.05, ** *p* < 0.01 versus NC group or ONC group.

**Figure 2 ijms-25-09978-f002:**
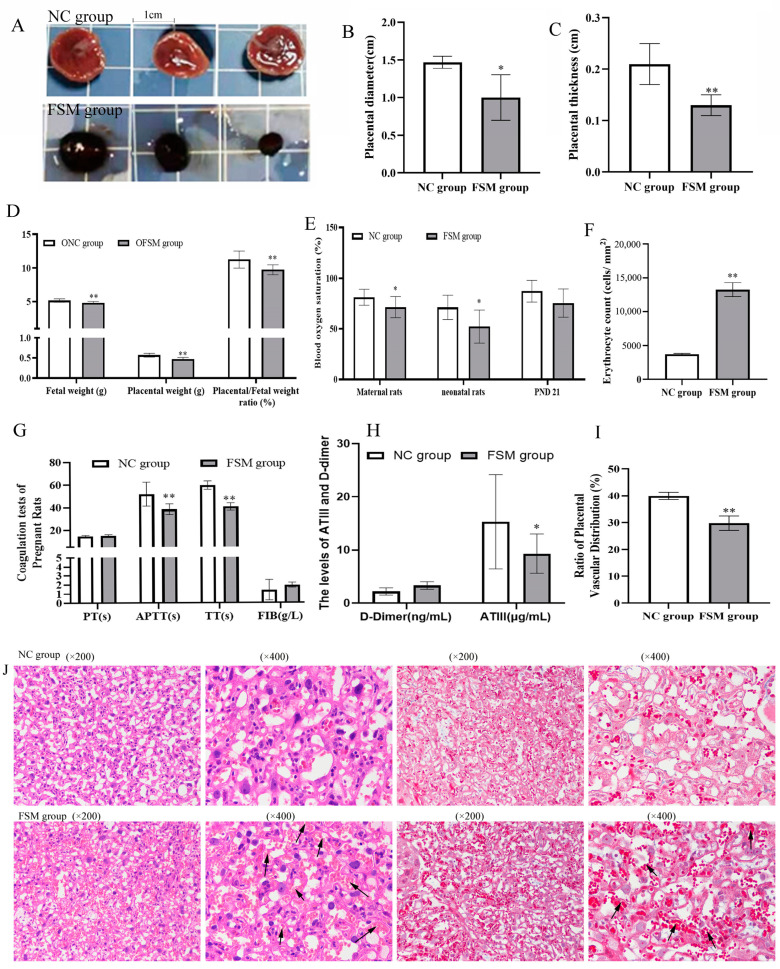
Effects of prenatal fear stress on placental efficiency, coagulatory biomarkers, and placental histopathology in pregnant rats. (**A**) the morphological features of placental tissues; (**B**) the diameters and (**C**) the thicknesses of placentas; (**D**) the weights of the fetus and placentas, and the placental efficiency; (**E**) the blood oxygen saturations of pregnant rats and their pups; (**F**) quantification of red blood cells in placental blood vessels using imageJ; (**G**,**H**) the coagulatory biomarkers of pregnant rats using the detection kits; (**I**) Quantitative analysis of the vascular network distribution in placental tissue using imageJ; (**J**) the H&E staining and Masson staining of the placentas were performed to evaluate the pathological changes of the placenta. The arrow indicates the presence of red blood cells within the blood vessels. NC group, normal control group; FSM group, fear stress model group; OFSM, offspring from the FSM group; ONC, offspring from the normal control group; PND 21, postnatal days of 21, PT, prothrombin time; APTT, activated partial thromboplastin time; TT, thrombin time; FIB, fibrinogen; ATIII, antithrombin III. Values are expressed as means ± standard deviations.* *p* < 0.05, ** *p* < 0.01 versus NC group or ONC group.

**Figure 3 ijms-25-09978-f003:**
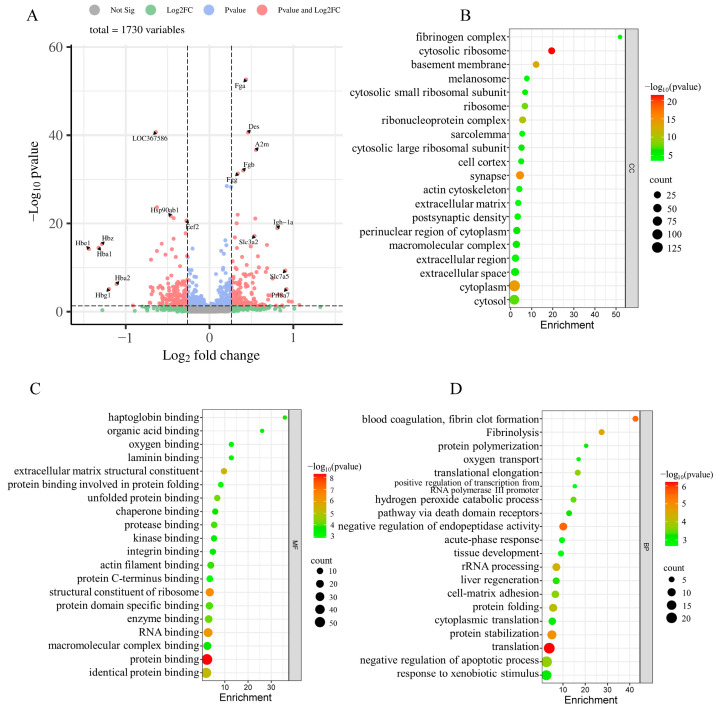
The volcano map of the identified proteins and GO classification of DEPs were shown. (**A**) The volcano map of the identified proteins was plotted based on its logarithmic fold change and *p*-value. *x*-axis represents the log2-fold change of protein expressions in FSM group compared to the NC group; The *y*-axis corresponds to the *p*-value of this fold change. Red dots: significant difference proteins; blue dots: proteins with fold changes > 0.83 or <1.2, and *p*-value < 0.05; grey and green dots: proteins with no significant difference in expression. (**B**–**D**) the GO classification of DEPs; (**B**) distributions of the DEPs for cellular components; (**C**) distributions of the DEPs for molecular functions; (**D**) distributions of the DEPs for biological processes. Only the top 20 Gene Ontology (GO) terms with significant differences (*p*-value < 0.05) are shown.

**Figure 4 ijms-25-09978-f004:**
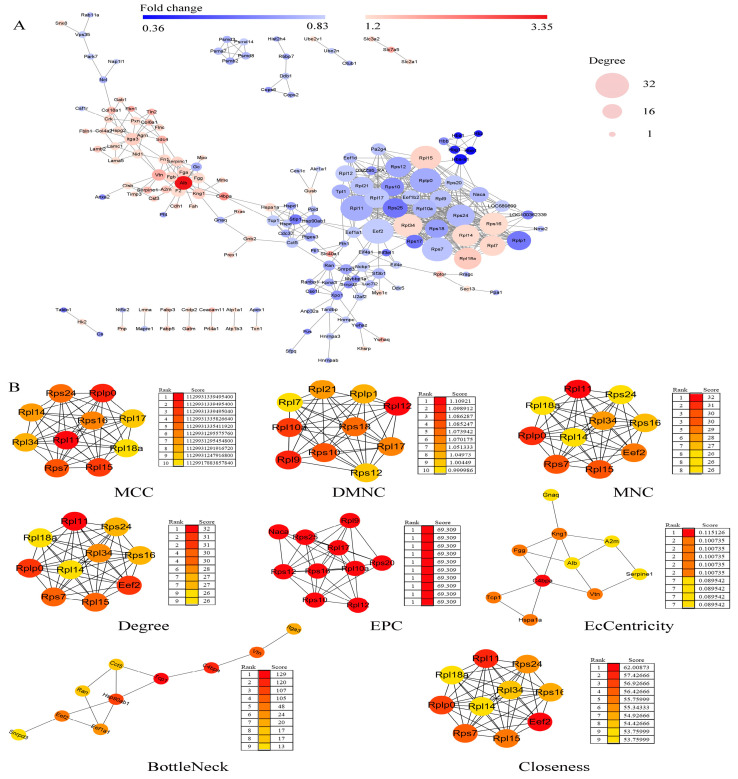
PPI network construction and hub genes extraction of DEPs. (**A**) The PPI network of the DEPs is shown. Network nodes represent proteins; edges represent protein–protein associations. Network analysis was set at high confidence (STRING score = 0.4) and high FDR stringency (0.01). Red indicates significantly increased; green indicates significantly decreased; width represents the degree of the interactions. (**B**) The top 10 hub genes were extracted with cytoHubba plug-in of Cytoscape software, and the eight algorithms are shown. Network nodes represent proteins; edges represent protein–protein associations. Color represents the score. MCC, Matthews Correlation Coefficient; DMNC, Neighborhood Component Centrality; MNC, Maximum Neighbor Connectivity; EPC, Edge Percolated Component.

**Figure 5 ijms-25-09978-f005:**
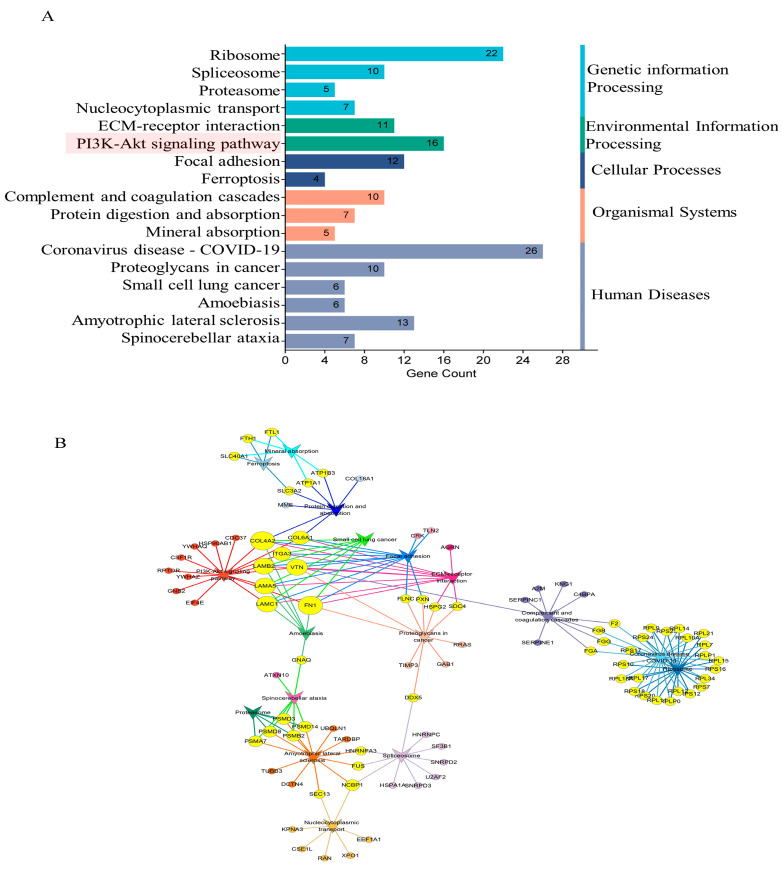
The results of KEGG enrichment of DEPs. (**A**) The significantly enriched pathways (*p*-value < 0.05). (**B**) The significant interactions of the pathways (*p*-value < 0.05). Circle-nodes represent DEPs. Arrow-shaped nodes represent the pathways. Edges represent connections between the nodes. Node color represents the pathways to which they belong. Yellow coloring represents the multiple pathways to which the proteins belong.

**Figure 6 ijms-25-09978-f006:**
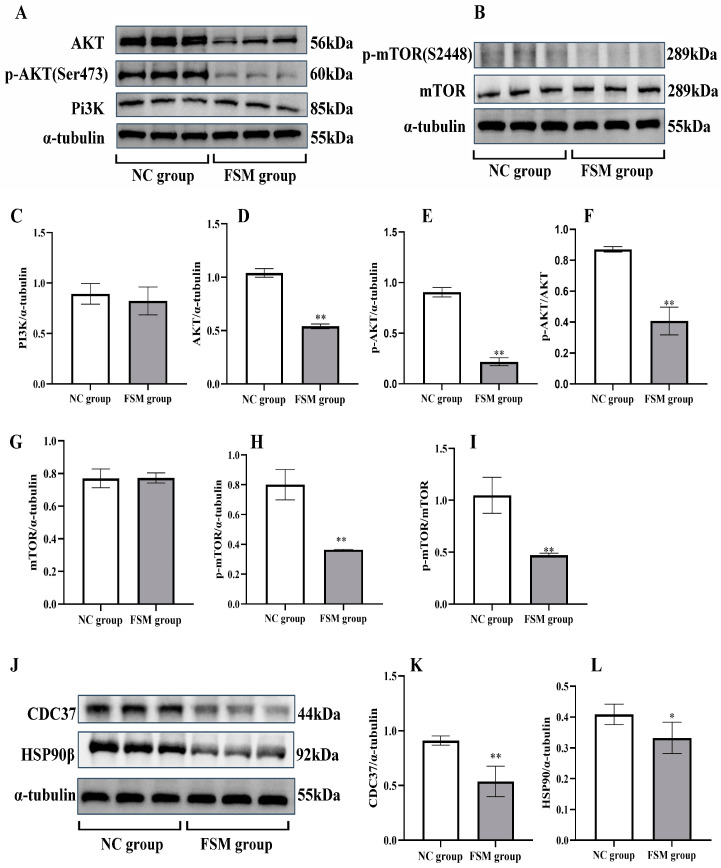
The expressions of PI3K/AKT/mTOR pathway in the placenta of rats detected by western blotting assay. (**A**) Representative western blotting images of PI3K, AKT, and p-AKT; (**B**) Representative western blotting images of mTOR and p-mTOR. The protein expressions of PI3K (**C**), AKT (**D**), and p-AKT (**E**) determined by image J software (1.53n); (**F**) The p-AKT/AKT ratio calculated. The protein expressions of mTOR (**G**) and p-mTOR (**H**) determined by image J software (1.53n); (**I**) The p-mTOR/mTOR ratio calculated. (J) Representative western blotting images of CDC37 and HSP90β. The protein expressions of CDC37 (**K**), and HSP90β (**L**) determined by image J software (1.53n). Results presented as means ± SD (*n* = 6). ** *p* < 0.01 and * *p* < 0.05 versus ONC group.

**Figure 7 ijms-25-09978-f007:**
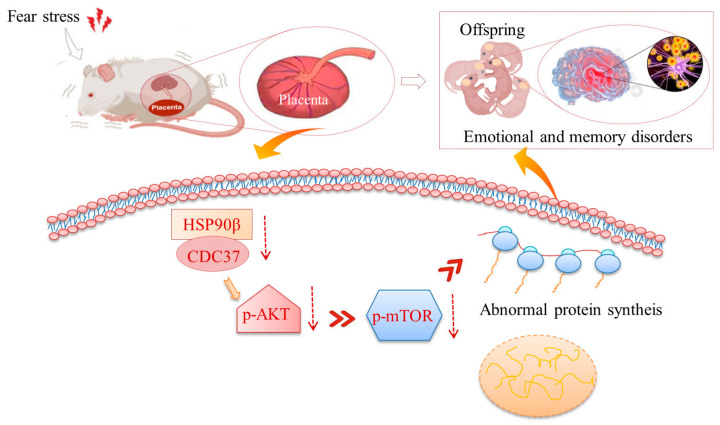
Schematic diagrams for prenatal stress-induced placental damage. Prenatal fear stress induces down-regulation of PI3K/AKT/mTOR pathway in placenta. These abnormalities in protein expression might damage the barrier function of the placenta and thereby cause abnormal development in the offspring.

**Figure 8 ijms-25-09978-f008:**
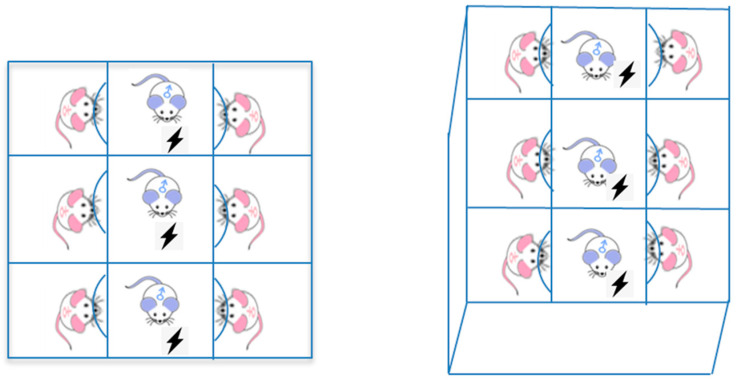
Fear stress schema illustration.

**Figure 9 ijms-25-09978-f009:**
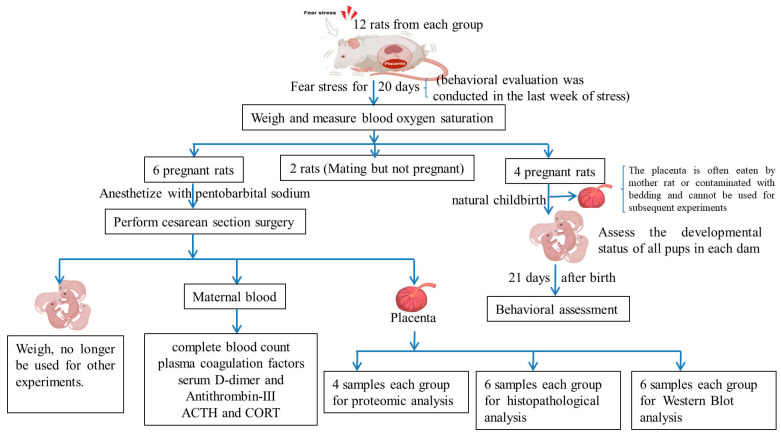
Schematic diagrams for the experimental protocol.

**Table 1 ijms-25-09978-t001:** The complete blood cell counts (CBC) of pregnant rats were determined.

Index	NC Group	FSM Group
White blood cells (×10^9^/L)	7.20 ± 2.92	3.48 ± 1.45 *
Lymphocyte (×10^9^/L)	3.87 ± 1.02	2.12 ± 0.79 **
Monocyte (×10^9^/L)	0.22 ± 0.12	0.17 ± 0.10
Neutrophil (×10^9^/L)	3.12 ± 1.82	1.20 ± 0.59 *
Percentage of lymphocytes (%)	55.70 ± 7.86	62.23 ± 6.54
Percentage of monocytes (%)	3.42 ± 0.45	4.55 ± 1.22
Percentage of neutrophils (%)	40.88 ± 7.67	33.22 ± 5.49
Red blood cell (×10^12^/L)	5.48 ± 0.27	5.70 ± 0.77
Hemoglobin (g/L)	107.00 ± 5.51	111.00 ± 13.67
Hematocrit (%)	33.33 ± 1.84	32.22 ± 3.62
Mean corpuscular volume (fL)	60.97 ± 1.44	56.80 ± 2.27 **
Mean corpuscular hemoglobin (pg)	19.50 ± 0.44	19.47 ± 0.46
Red cell distribution width (%)	12.32 ± 0.71	13.08 ± 1.41
Platelet (×10^9^/L)	940.50 ± 128.04	1421.00 ± 275.97 **
Mean platelet volume (fL)	5.93 ± 0.24	8.07 ± 0.68 **

Note: * *p* < 0.05, ** *p* < 0.01 versus normal group.

**Table 2 ijms-25-09978-t002:** Top 10 up- and down-regulated proteins in the placenta were show.

Gene Names	Protein Names	Fold Change	*p*-Value
Alb	Serum albumin	3.344481605	0.00107535
C4bpa	C4b-binding protein alpha chain	2.100840336	0.028742183
Plp2	Proteolipid protein 2	1.976284585	0.00518908
Slc40a1	Protein LOC100911874	1.923076923	0.004230614
Prl8a7	Prolactin-8A7	1.890359168	7.2348 × 10^−6^
Slc7a5	Large neutral amino acids transporter small subunit 1	1.872659176	4.26129 × 10^−10^
Fbn1	Fibrillin 1	1.824817518	0.035886107
Aqp1	Aquaporin-1	1.814882033	0.000190706
Vtn	Protein Vtn	1.76366843	0.000118123
Igh-1a	Ig gamma-2B chain C region	1.751313485	1.1581 × 10^−19^
Rplp1	60S acidic ribosomal protein P1	0.619195046	0.000108538
Rps17	40S ribosomal protein S17	0.616522811	1.57927 × 10^−12^
Rps25	40S ribosomal protein S25	0.61462815	2.51193 × 10^−6^
Eif3e	Eukaryotic translation initiation factor 3 subunit E	0.612369871	0.010033231
Taldo1	Transaldolase	0.609756098	0.008266795
Hba2	Protein Hba2	0.463606861	5.45252 × 10^−7^
Hbg1	Beta-globin chain (Fragment)	0.43554007	7.10019 × 10^−6^
Hbz	Protein Hbz	0.410677618	4.77399 × 10^−16^
Hba1	Hemoglobin subunit alpha-1/2	0.399361022	3.2938 × 10^−15^

**Table 3 ijms-25-09978-t003:** Peptide segment labelling information.

	Sample Name	Labels Information
NC group	N1	113
	N2	114
	N3	115
	N4	116
FSM group	M1	117
	M2	118
	M3	119
	M4	121

## Data Availability

The original contributions presented in the study are included in the article/Appendix A; further inquiries can be directed to the corresponding author/s.

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
