# Peer review of "ITRAQ Based Proteomics Reveals the Potential Mechanism of Placental Injury Induced by Prenatal Stress"

_ijms, 2024, doi:10.3390/ijms25189978_

Round 1
Reviewer 1 Report
Comments and Suggestions for Authors
Li and colleagues present data of an interesting study aimed at identifying pathophysiological changes in the placenta due to fear prenatal fear induced stress. The authors are commended for not limiting their study with proteomics and identification of proteins that are up- and down-regulated, and including data regarding behavioral changes in mom and the offspring, as well as including placental histology and maternal stress hormones near term. Overall the authors report prenatal fear induced stress inhibition of placental expression of AKT/mTOR, reduced neonatal oxygenation and placental weight, and induced anxiety and depression-like behaviors. Overall this is an interesting study, however a number of concerns need to be addressed.
-) Of great importance is the fact that data regarding the placenta, neonates and offspring need to be separated by biological sex. Female and male sex differences in responses to maternal stress have been extensively reported.
-) at what time of day was blood collected for ACTH and CORT analysis? Was this done simultaneously for all animals?
-) what were the litter sizes?
-) regarding Figure 2E: What area of the placenta is highlighted? What are the arrows pointing at? If increased number of erythrocytes are indicated, please quantify. Importantly, please expand on the idea that increased numbers of erythrocytes correspond to vascular changes in the placenta and/or a "pathological hypercoagulable state" in mom. Were any morphological changes observed considering placental weight is reduced? Are there changes in some of the surface area of the placentas? Are there changes in vascular areas?
-) Figure 3A; this figure can go as a supplement. Instead, include a table with the top 10 up- and down-regulated proteins.
-) Figure 3B; please indicate what "MCC", DMNC" "MNC" etc means.
-) Considering prenatal fear induced stress is connected to increased ACTH and CORT in mom, how do the author speculate this leads to placental changes? How many proteins identified are regulated by glucocorticoid receptor-mediated transcription/translation (i.e., signaling)?
Comments on the Quality of English LanguageNo concerns are noted.
Author Response
Comments 1: [Of great importance is the fact that data regarding the placenta, neonates and offspring need to be separated by biological sex. Female and male sex differences in responses to maternal stress have been extensively reported.]
Response 1: [Thank you for your valuable comments. We apologize for not analyzing offspring gender as an influencing factor. Due to the small size of newborn mice, distinguishing gender was challenging, and thus, we did not pay attention to the gender of the offspring rat corresponding to each placental sample. We have reviewed recent studies on the effects of stress during pregnancy on offspring, noting many that also did not focus on offspring gender (see table below). Additionally, our study aimed to explore the impact of fear-based stress during pregnancy on placental structure and function. The behavioral analysis of pregnant rat and their offspring was primarily conducted to ensure the model's reproducibility and stability, thereby enhancing the reliability of our results. Consequently, we did not prioritize gender as an influencing factor. We greatly appreciate your feedback and will include an analysis of the pups' gender in our future research. Thank you once again for your suggestions.]
|
Title |
Journal |
Impact factor |
Published year |
Species |
Model |
analyses |
|
Fetal CCL2 signaling mediates offspring social behavior and recapitulates effects of prenatal stress |
Brain Behav Immun. |
8.8 |
2024.1 |
mice |
Maternal stress |
Failure to consider the influence of gender factors |
|
The effect of prenatal stress on offspring depression susceptibility in relation to the gut microbiome and metabolome |
J Affect Disord. |
4.9 |
2023.10 |
Rat |
Prenatal stress rat model |
Failure to consider the influence of gender factors |
|
PM2.5 leads to adverse pregnancy outcomes by inducing trophoblast oxidative stress and mitochondrial apoptosis via KLF9/CYP1A1 transcriptional axis |
Elife. |
6.4 |
2023.9 |
mice |
PM2.5-exposed pregnant mouse model |
Failure to consider the influence of gender factors |
|
Chlorpyrifos induces placental oxidative stress and barrier dysfunction by inducing mitochondrial apoptosis through the ERK/MAPK signaling pathway: In vitro and in vivo studies |
Sci Total Environ. |
8.2 |
2023.9 |
mice |
Exposure to Chlorpyrifos during Pregnancy |
Failure to consider the influence of gender factors |
Comments 2:[ at what time of day was blood collected for ACTH and CORT analysis? Was this done simultaneously for all animals?]
Response 2: [Thanks for your careful reading and useful comments. We apologize for not providing a clear explanation of the entire experimental process. We have supplemented it in article and marked in red (page 16, line 19-21). (1) Blood collection was exclusively focused on pregnant rats. As this study mainly focuses on the impact of stress during pregnance on the placenta, we primarily assessed the blood parameters of pregnant rats. Blood collection and parameter testing were not performed on their offspring, female rats that did not conceive, or the male rats used in the experiment. Thus, blood collection was exclusively targeted at the pregnant rats. (2) We have included the specific blood collection time and process for pregnant mice in the manuscript (Page 16, line 19-21).]
Comments 3:[ what were the litter sizes?]
Response 3: [Thanks for your careful reading and useful comments. We're sorry for neglecting the indicator of litter size. We have supplemented it in article and marked in red. (Page 3, Figure 1I, line 11, 22, 23, and Page 16, line24-27)]
Comments 4:[ regarding Figure 2E: What area of the placenta is highlighted? What are the arrows pointing at? If increased numbers of erythrocytes are indicated, please quantify. Importantly, please expand on the idea that increased numbers of erythrocytes correspond to vascular changes in the placenta and/or a "pathological hypercoagulable state" in mom. Were any morphological changes observed considering placental weight is reduced? Are there changes in some of the surface area of the placentas? Are there changes in vascular areas?]
Response 4: [Thanks for your comments on this article. Your comments on the sampling location and morphological features of the placenta are critical and essential. We have supplemented the specific sampling areas of the placenta according to your suggestion (Page 16, lines 50-51), and provided a detailed description corresponding to the areas indicated by the arrows (Page 5, line10, and page 6, line 1). Additionally, we have meticulously quantified both the placental vascular area and the count of red blood cells within the vessels according to your suggestion (Page 5, Fig. 2I and 2F, and Page 6, line 7-11). Furthermore, we have supplemented representative images that depict the morphological alterations of the placenta (Page 5, Fig. 2A, and Page 6, line 7-11), and provided supplementary data on variations in placental diameter and thickness (Page 5, Fig. 2B and 2C, Page 6, line 7-11). We are immensely grateful for your insightful suggestions, which have significantly enhanced the rigor and scientific integrity of our manuscript.]
Comments 5:[ Figure 3A; this figure can go as a supplement. Instead, include a table with the top 10 up- and down-regulated proteins.]
Response 5: [We are appreciated of your careful reading and useful comments. We have supplemented the first 10 upregulated and downregulated proteins according to your suggestion (Page 6, line 31-32, and Table 2). Additionally, since protein interactions are a commonly used analytical method, this graph is essential for the subsequent screening of key proteins. Therefore, we feel that retaining this result would be better. Thank you again for your suggestion.]
Comments 6:[ Figure 3B; please indicate what "MCC", DMNC" "MNC" etc means.]
Response 6: [Thanks for your careful reading and useful comments. We have annotated "MCC", "DMNC", "MNC", etc. according to your suggestion (Page 7, line 20, and page 8, line 1-2 ).]
Comments 7: [ Considering prenatal fear induced stress is connected to increased ACTH and CORT in mom, how do the author speculate this leads to placental changes? How many proteins identified are regulated by glucocorticoid receptor-mediated transcription/translation (i.e., signaling)?]
Response 7: [Special thanks to you for your insightful comments. (1) Our prior research has revealed that stress induced by prenatal fear is associated with elevated levels of maternal ACTH and CORT, which in turn trigger epigenetic modifications in the placenta. We have incorporated our prior findings into the discussion section (Page 11, lines 22-26). However, the specific changes in protein expression resulting from these epigenetic modifications remain unclear. Consequently, we have systematically evaluated differential proteins using proteomics. The inclusion of these additions enhances the rigor of our research. We extend our gratitude once again for your suggestions. (2) Through in-depth analysis of protein interaction results, we identified 14 proteins that can interact with glucocorticoid receptors. This suggests that these proteins may be regulated by glucocorticoid receptor-mediated signaling pathways. We have expanded on this in our discussion (Page 13, lines 12-15). Your feedback has enhanced the richness and scientific validity of our paper. We greatly appreciate your suggestions.]

Reviewer 2 Report
Comments and Suggestions for Authors
The current manuscript explores the relationship between prenatal stress and placental injury in pregnant rats through the use of proteomic labeling and quantification via the iTRAQ method. The study is generally intriguing, but it would benefit from a more comprehensive revision and improved structuring of its subsections. I have listed a few comments that may help to improve the manuscript.
1. The authors performed the iTRAQ labeling and only used two labels for the NC and FSM groups. In contrast, they selected 24 rats for the study 12 in each group. It is not clear why the authors used iTRAQ 8 plex for 24 samples and used only two labels.
2. Section 2.5. iTRAQ-based proteomic analyses need to be rewritten with more details. What is uni_rattus_10114 database? What proteome was used? List out the total proteins in each group.
3. The iTRAQ quantification is ambiguous and needs more details in the methods section.
4. In Figure 4A, the authors stated they used “fold change criterion of 1.2 (upregulation ratio >1.2 or downregulation ratio <0.83)…” but in the volcano plot it does not appear that 1.2 FC was applied. The Figure need to be updated.
5. Protein-Protein interaction (PPI) network construction and hub genes extraction section also need more details. What FDR was used? In Figure 3 please provide the color score bar.
6. What criteria were used for the KEGG analysis?
7. Page 13, line 427-428 “In conclusion, we employed LC-MS based metabonomics and iTRAQ based proteomics to ascertain if fear stress in pregnant rats has an impact on placental metabonomics.” In the current manuscript, the authors did not use metabolomics.
Author Response
Comments 1: [The authors performed the iTRAQ labeling and only used two labels for the NC and FSM groups. In contrast, they selected 24 rats for the study 12 in each group. It is not clear why the authors used iTRAQ 8 plex for 24 samples and used only two labels.]
Response 1: [We sincerely appreciate your comment, which has enhanced the accuracy of our paper's methodology. Our proteomic analysis was conducted by BGI Protein, with 4 samples in each group, totaling 8 samples. We regret overlooking the labeling issue during the translation of the paper, which may have arisen from our limited familiarity with the specific detection process. Thank you for your thorough review. Upon reviewing the detection report, we discovered that our normal group was labeled with 113, 114, 115, 116, and the model group was labeled with 117, 118, 119, and 121, respectively. We have made revisions to the methods (Page 17, lines 28-30). Thank you again for identifying this issue.]
Comments 2: [ Section 2.5. iTRAQ-based proteomic analyses need to be rewritten with more details. What is uni_rattus_10114 database? What proteome was used? List out the total proteins in each group.]
Response 2: [Thanks for your careful reading and useful comments. We have updated the description and provided the total protein identified and outlined the screening methods and criteria (Page 6, lines 23-31). Additionally, we have corrected misinformation, such as the uni_rattus_10114 database was specifically constructed for this experiment rather than for protein type matching. Proteins were primarily identified by matching against the NCBI and UniPort databases. Thank you once again for your suggestions.]
Comments 3: [The iTRAQ quantification is ambiguous and needs more details in the methods section.]
Response 3: [Thank you for your valuable feedback on this article. Your insights regarding the iTRAQ quantification method are of utmost importance. We have supplemented the detailed methods for iTRAQ quantification (Page 18, line 6-18).]
Comments 4: [In Figure 4A, the authors stated they used “fold change criterion of 1.2 (upregulation ratio >1.2 or downregulation ratio <0.83)…” but in the volcano plot it does not appear that 1.2 FC was applied. The Figure need to be updated.]
Response 4: [Thank you very much for your suggestion. We have redone the volcano plot (Page 8, fig. 4A).]
Comments 5: [Protein-Protein interaction (PPI) network construction and hub genes extraction section also need more details. What FDR was used? In Figure 3 please provide the color score bar.]
Response 5: [Thank you very much for your suggestion. We have supplemented the parameter settings for protein-protein interaction (PPI) network construction (Page 7, line 13) and included a color score bar for Figure 3 (Page 7, Fig. 3).]
Comments 6: [What criteria were used for the KEGG analysis?]
Response 6: [Thank you for your comment. We have added the criteria used for KEGG analysis (Page 9, line 7-8)]
Comments 7: [Page 13, line 427-428 “In conclusion, we employed LC-MS based metabonomics and iTRAQ based proteomics to ascertain if fear stress in pregnant rats has an impact on placental metabonomics.” In the current manuscript, the authors did not use metabolomics. Did not use metabolomics.]
Response 7: [Thank you very much for your comment. We deeply apologize for the description error that occurred this time. We have made the necessary corrections (Page 14, line 27-28).]

Round 2
Reviewer 1 Report
Comments and Suggestions for Authors
The revised manuscript is very much appreciated and most of my previous comments have been addressed.
Regarding my main concern about not including (i.e., separating the data) biological sex as a variable. First, the authors should refrain from using the term gender. We do not know the gender of the animals, but we can establish the biological sex of the animal (XY males, and XX females). Second, although visible assignment of biological sex can be difficult in neonates, this usually is readily done by PCR genotyping. [Dhakal P, Soares MJ. Single-step PCR-based genetic sex determination of rat tissues and cells. Biotechniques. 2017 May 1;62(5):232-233. doi: 10.2144/000114548. PMID: 28528577]. Finally, highlighting published studies that did not include biological sex in their data analysis can not be a justification for not including this important variable.
The only minor issue remaining relates to comment 7 (GR-targets identified). It is very much appreciated that analysis reveals 14 protein targets that are regulated to GR signaling. Please mention/highlight which 14 proteins these are (it is not clear where they are listed).
Author Response
Comments 1: [Regarding my main concern about not including (i.e., separating the data) biological sex as a variable. First, the authors should refrain from using the term gender. We do not know the gender of the animals, but we can establish the biological sex of the animal (XY males, and XX females). Second, although visible assignment of biological sex can be difficult in neonates, this usually is readily done by PCR genotyping. [Dhakal P, Soares MJ. Single-step PCR-based genetic sex determination of rat tissues and cells. Biotechniques. 2017 May 1;62(5):232-233. doi: 10.2144/000114548. PMID: 28528577]. Finally, highlighting published studies that did not include biological sex in their data analysis can not be a justification for not including this important variable.]
Response 1: [Thank you very much for your suggestion. In our examination of the offspring's behavior, we adopted the criterion of sex parity in distribution (equal distribution of males and females, Page 15, lines 18-19). In addition, to consider the influence of sex factors, we used Two-Way ANOVA to statistically analyze the offspring's behavioral detection results. We have supplemented this in the manuscript and marked it in blue (Page 4, lines 22 - 25). Thank you again for your suggestions.]
Comments 2: [The only minor issue remaining relates to comment 7 (GR-targets identified). It is very much appreciated that analysis reveals 14 protein targets that are regulated to GR signaling. Please mention/highlight which 14 proteins these are (it is not clear where they are listed).]
Response 2: [We are greatly appreciative of your insightful recommendations. We have added protein targets regulated by GR signal in the manuscript according to your suggestion (Page 13, lines 13 - 16). In compliance with another expert's recommendation, we modified the confidence threshold for the protein interaction network to 0.7, a change that has yielded a more reliable dataset. Post-adjustment, it was observed that two proteins are capable of interacting with NR3C1 (Name of glucocorticoid receptor gene). The requisite alterations have been incorporated into the manuscript, with the names of the genes identified included on Page 13, lines 13-16. We appreciate your insightful advice.]

Reviewer 2 Report
Comments and Suggestions for Authors
The revised version of the manuscript looks much improved. However, the authors still need to revise the manuscript.
comments 3: [The iTRAQ quantification is ambiguous and needs more details in the methods section.] The authors need to provide the quantification workflow, and which samples and controls were used, how they were normalized.
comments 4: [In Figure 4A, the authors stated they used “fold change criterion of 1.2 (upregulation ratio >1.2 or downregulation ratio <0.83)…” but in the volcano plot it does not appear that 1.2 FC was applied. The revised figure is still not showing the cutoff of 1.2 FC. The vertical line is still close to 0 and not at 1.
Comments 5: [Protein-Protein interaction (PPI) network construction and hub genes extraction section also need more details. What FDR was used? In Figure 3 please provide the color score bar.] The authors used medium confidence which yields a lot more connections. The authors should use high confidence (0.7) and high FDR stringency to minimize the network.
Author Response
Comments 3: [The iTRAQ quantification is ambiguous and needs more details in the methods section. The authors need to provide the quantification workflow, and which samples and controls were used, how they were normalized.]
Response 3: [We are profoundly grateful for your invaluable suggestion. Consequently, we have incorporated the quantification protocol, detailed the specific samples and controls employed, as well as delineated the normalization strategy utilized for iTRAQ (Page 18, lines 18-23), which have been marked in blue in the paper.]
Comments 4: [In Figure 4A, the authors stated they used “fold change criterion of 1.2 (upregulation ratio >1.2 or downregulation ratio <0.83)…” but in the volcano plot it does not appear that 1.2 FC was applied. The revised figure is still not showing the cutoff of 1.2 FC. The vertical line is still close to 0 and not at 1.]
Response 4: [Thank you very much for your comment. Our screening criteria are upregulation ratio > 1.2 or downregulation ratio < 0.83. The horizontal axis of the volcano plot is log2FC. Thus, the positions of the left and right vertical lines are log20.83=-0.268 and log21.2=0.263 respectively. Therefore, the vertical lines are relatively close to 0.]
Comments 5: [Protein-Protein interaction (PPI) network construction and hub genes extraction section also need more details. What FDR was used? In Figure 3 please provide the color score bar. The authors used medium confidence which yields a lot more connections. The authors should use high confidence (0.7) and high FDR stringency to minimize the network.]
Response 5: [We are greatly appreciative of your insightful recommendations. We have added more detailed descriptions to the protein-protein interaction (PPI) network construction and hub gene extraction section (Page 7, lines 4-11). Furthermore, we have redrawn the protein-protein interaction (PPI) network diagram with high confidence (0.7) and FDR stringency (0.01) (Page 7, lines 16), followed by a re-evaluation of the hub genes, now complemented with a color-coded score bar (Page 7, Figure 3). Your suggestions have been invaluable.]

Round 3
Reviewer 1 Report
Comments and Suggestions for Authors
Thank you for the resubmission and addressing the two remaining issues regarding inclusion of sex as a biological variable in the study and the GR targets.
I have major concerns regarding the previous response to the important issue of biological sex. First, biological sex was not included and it was then indicated that this was difficult to do in neonates, which I completely agree with. As such I recommended published PCR genotyping assays that could accurately do this on frozen rat tissues. However, in this second resubmission, animals were assigned to groups that now include 5 males and 5 females?? How exactly was this done? This is not clear at all. What is the "criteria of sex parity in distribution"? Were animals assigned to seach group AFTER experiments were completed? If animals were genotyped, please provide all the corresponding data in the material and methods and gel electropheresis images as supplemental material. If PCR genotyping was not done; how were these animals assigned as male and female in each group before experiments were done??
Author Response
Comments 1: [I have major concerns regarding the previous response to the important issue of biological sex. First, biological sex was not included and it was then indicated that this was difficult to do in neonates, which I completely agree with. As such I recommended published PCR genotyping assays that could accurately do this on frozen rat tissues. However, in this second resubmission, animals were assigned to groups that now include 5 males and 5 females?? How exactly was this done? This is not clear at all. What is the "criteria of sex parity in distribution"? Were animals assigned to seach group AFTER experiments were completed? If animals were genotyped, please provide all the corresponding data in the material and methods and gel electropheresis images as supplemental material. If PCR genotyping was not done; how were these animals assigned as male and female in each group before experiments were done??]
Response 1: [(1) We are profoundly grateful for your critical feedback on our manuscript. We extend our sincere apologies for the lack of clarity in the description of our experimental procedures, as well as for the inaccuracies present in our initial response(criteria of sex parity in distribution) . We have revised the method, and included supplementary details, which have marked in blue (page 15, line 13-24). Moreover, to facilitate a comprehensive understanding, we have drawn a schematic flowchart illustrating the experimental protocol, which is depicted in the the attached file.
(2) Biochemical analyses were conducted on tissue specimens procured from rats that underwent cesarean deliveries, with each group consisting of six rats (page 16, line 24-37). The neonatal rats were exclusively evaluated for their body weight and number of litters. These neonates were not reared or subjected to subsequent behavioral assessments to mitigate the risk that the surgical procedure and the anesthetics employed might influence the outcomes of future behavioral tests.
(3) Subsequent to natural parturition (4 pregnant rats in each group), the dams and their progeny were raised normally and reserved for subsequent behavioral assessments (pages 15, line 13-16). The evaluation of offspring mainly includes behavioral and developmental assessment. All of the offspring rats need to undergo developmental assessment (Not considering the influence of sex factors. pages 15, line 16-20). Behavioral assessment were conducted between 21-27 days after the offspring were born (pages 15, line 20-24), at which period the offspring had attained a sizeable stature, and could be raised separately from their mothers, making it easy to distinguish sex. Consequently, during the behavioral evaluations, 10 rats (5 male and 5 female) from each group were collected for the experiment. We investigate the behavior of the offspring mainly to ensure the stability of the modeling method, so we evaluate the behavior of the offspring in each batch of experiments. Thank you again for your suggestions.
(4)Thank you sincerely for providing us with the PCR genotyping method. This method is simple and convenient to operate, and we would like to express our gratitude for your serious comments and suggestions. Initially, when designing the experiment, we speculated that stress during pregnancy may have a certain impact on the function of the placenta. Regardless of whether the offspring are female or male, the damage caused by stress during pregnancy to placental function may exist. However, the impact of this damage on female and male offspring may be different. As the focus of this study was on the impact of stress during pregnancy on the placenta, we did not pay special attention to sex factors at that time, which resulted in us not being able to detect the corresponding biological sex of the placental samples in a timely manner before testing proteomics. Our proteomics result was detected by Beijing Huada Protein Research and Development Center Co., Ltd. During the testing process, and approximately 200mg of tissue samples were required. As each placental sample is not very heavy, the sample weight of the model group will be slightly smaller, which resulted in us not retaining the placental samples used for proteomics testing at that time. Therefore, we are unable to supplement the sex corresponding to the placental samples sent for testing. We are sorry that we did not consider this in our early experimental design. We also have many placental samples stored in the refrigerator, but they are all independent placental samples, and their sex does not correspond to the samples used for proteomic testing. Therefore, we are very sorry that we cannot supplement the biological sex to the placental samples. Thank you very much for your suggestions and methods. In our future research, we will further investigate the influence of sex factors. Thank you again for your suggestion.]

Reviewer 2 Report
Comments and Suggestions for Authors
In the revised version, the authors made minor additions that did not really help to improve the manuscript.
The authors' comment from the third round still hasn't been addressed. It would be helpful if the authors could provide a separate table that clearly defines the samples and controls.
The authors' response to comment 4 regarding the log2 FC is misguided. The log2 value should be greater than 2, and the line should be drawn at a minimum of 1. When using the author's calculations, if they set the log2 cutoff value to 1, then all proteins would be considered significant. However, this approach is not applicable in biomarker discovery using proteomics. Instead, the log2 FC should be a minimum of 1 (with the line at 1).
Author Response
Comments 1: [The authors' comment from the third round still hasn't been addressed. It would be helpful if the authors could provide a separate table that clearly defines the samples and controls.]
Response 1: [Thank you very much for your comments. We have added a table containing the groups, samples, and corresponding labels according to your suggestions (page 18, table 3). Furthermore, our team engaged in dialogue with the company's technical staff. Given that iTRAQ is a quantitative labeling technique, the labeled samples are mixed prior to being introduced into the analytical instrument. Consequently, only a single identification table is generated, rather than separate tables for individual sample sets. Thank you again for your comment.]
Comments 2: [ The authors' response to comment 4 regarding the log2 FC is misguided. The log2 value should be greater than 2, and the line should be drawn at a minimum of 1. When using the author's calculations, if they set the log2 cutoff value to 1, then all proteins would be considered significant. However, this approach is not applicable in biomarker discovery using proteomics. Instead, the log2 FC should be a minimum of 1 (with the line at 1).]
Response 1: [(1) Thank you again for your comments. We attempted to set 2 as the cutoff value, with an upregulation ratio>2 or a downregulation ratio<0.5. After drawing, we found that only 7 differentially expressed proteins remained, among which 5 were downregulated and 2 were upregulated (see figure in the attached file). This setting is not conducive to our attention to the extensive changes in placental proteomics after stress. Therefore, we consulted a large amount of literature [1-5] and ultimately decided to use fold changes>1.2 or<0.83 with p-value<0.05 as cutoff.
(2) Concurrently, we conducted an extensive review of the pertinent literature, revealing that a multitude of proteomics investigations adopt 1.2 as the threshold for the heatmap visualization[2-3, 6-7] (see the figures in the attached file). Additionally, certain researchers have opted for thresholds of 2 or 1.5. Nevertheless, based on our detection results, using fold changes>1.2 or<0.83 with p-value<0.05 as cutoff could yield more favorable outcomes. I am once again grateful for your insightful feedback.
Reference
- Zhou, X.; Sheikh, A. M.; Matsumoto, K. I.; Mitaki, S.; Shibly, A. Z.; Zhang, Y.; A, G.; Yano, S.; Nagai, A., iTRAQ-Based Proteomic Analysis of APP Transgenic Mouse Urine Exosomes. International journal of molecular sciences 2022, 24, (1).
- Zhang, L. Y.; Huang, T. T.; Li, L. P.; Liu, D. P.; Luo, Y.; Lu, W.; Huang, N.; Ma, P. P.; Liu, Y. Q.; Zhang, P.; Yang, B. C., ITRAQ-based proteomics analysis of human ectopic endometrial stromal cells treated by Maqian essential oil. BMC complementary medicine and therapies 2023, 23, (1), 427.
- Li, R.; Chen, M.; Yan, D.; Chen, L.; Lin, M.; Deng, B.; Zhuang, L.; Gao, F.; Leung, G. P.; You, J., iTRAQ-based quantitative proteomics revealing the therapeutic mechanism of a medicinal and edible formula YH0618 in reducing doxorubicin-induced alopecia by targeting keratins and TGF-β/Smad3 pathway. Heliyon 2024, 10, (12), e33051.
- Zhang, M.; Wang, D.; Xu, X.; Xu, W.; Zhou, G., iTRAQ-based proteomic analysis of duck muscle related to lipid oxidation. Poultry science 2021, 100, (4), 101029.
- He, Y.; Lin, J.; Tang, J.; Yu, Z.; Ou, Q.; Lin, J., iTRAQ-based proteomic analysis of differentially expressed proteins in sera of seronegative and seropositive rheumatoid arthritis patients. Journal of clinical laboratory analysis 2022, 36, (1), e24133.
- Cheng, Y.; Liu, M.; Tang, H.; Chen, B.; Yang, G.; Zhao, W.; Cai, Y.; Shang, H., iTRAQ-Based Quantitative Proteomics Indicated Nrf2/OPTN-Mediated Mitophagy Inhibits NLRP3 Inflammasome Activation after Intracerebral Hemorrhage. Oxidative medicine and cellular longevity 2021, 2021, 6630281.
- Cheng, S.; Zhao, F.; Wen, L.; Yang, B.; Wang, X. Z.; Huang, S. N.; Jiang, X.; Zeng, W. B.; Sun, J. Y.; Zhang, F. K.; Shen, H. J.; Fortunato, E.; Luo, M. H.; Cheng, H., iTRAQ-Based Proteomics Analysis of Human Cytomegalovirus Latency and Reactivation in T98G Cells. Journal of virology 2022, 96, (2), e0147621.]

Round 4
Reviewer 1 Report
Comments and Suggestions for Authors
My previous comments have been addressed. Please ensure the figure illustrating the experimental procedure is provided in the final manuscript.
Author Response
Comments 1: [My previous comments have been addressed. Please ensure the figure illustrating the experimental procedure is provided in the final manuscript.]
Response 1: [We are greatly appreciative of your insightful recommendations. we have supplemented a figure to illustrate the experimental procedure, based on your comments and suggestions (page 15, figure 8). Your positive and constructive comments and suggestions on our manuscript have greatly improved its quality. Thank you again for your comments and suggestions.]

Reviewer 2 Report
Comments and Suggestions for Authors
The manuscript can be accepted.
Author Response
Comments 1: [The manuscript can be accepted.]
Response 1: [We greatly appreciate your agreement to accept our manuscript. On behalf of my co-authors, we greatly appreciate your comments on our manuscript. Your positive and constructive feedback and suggestions have greatly improved the quality of our manuscript. During the process of revising the manuscript, we have gained a deeper understanding of the parameter settings during the protein-protein interaction network drawing process. Thank you again for your comments and suggestions.]
